# Higher Education in Mexico: The Effects and Consequences of the COVID-19 Pandemic

**Guillermo M. Chans** [1,2] , **Angelica Orona-Navar** [3], **Carolina Orona-Navar** [3,4] **and Elvia P. Sánchez-Rodríguez** [5,*]

1   Institute for the Future of Education, Tecnologico de Monterrey, Monterrey 64849, Mexico; guillermo.chans@tec.mx
2   School of Engineering and Sciences, Tecnologico de Monterrey, Mexico City 01389, Mexico
3   School of Engineering and Sciences, Tecnologico de Monterrey, Monterrey 64849, Mexico; a00824057@tec.mx (A.O.-N.); carolina.orona@tec.mx (C.O.-N.)
4   Institute of Advanced Materials for Sustainable Manufacturing, Tecnologico de Monterrey, Monterrey 64849, Mexico
5   School of Engineering and Sciences, Tecnologico de Monterrey, Atizapan de Zaragoza 52926, Mexico
*   Correspondence: elvia.sanchez@tec.mx

**Abstract:** This review examines the impact of the COVID-19 pandemic on higher education in Mexico. It acknowledges the difficulties and disruptions caused by the global emergency but also emphasizes the opportunities for reflection and learning that have emerged. This work employed a comprehensive methodology, including a thorough literature search across multiple academic databases and consultation with national statistical sources and newspapers. Sixty-nine articles were selected based on predetermined keywords and criteria, leading to the identification of two central themes: impacts and implications on higher education and innovation in teaching and learning experiences. The findings shed light on the effects of the pandemic and highlight the need for resources, pedagogical considerations, and a reevaluation of priorities in the education sector. The review concludes by emphasizing the importance of improving equity, quality, and long-term sustainability in higher education in Mexico while recognizing the opportunity for educational reform in the post-pandemic era.

**Keywords:** higher education; Mexico; COVID-19; educational innovation; remote education; digital learning technologies; educational platforms

## 1. Introduction

The global community has recently been confronted with a major crisis, namely the COVID-19 pandemic. The severity of this critical situation stems mainly from the high number of fatalities and its impact on the economy, politics, and society. Since the first case was detected in China in December 2019, considering the unusually rapid spread of the virus worldwide, the World Health Organization (WHO) declared a public health emergency of international concern on 30 January 2020. It recognized it as a pandemic on 11 March 2020 [1].

Although having a head start over Asian and Western European countries struck by the initial COVID-19 outbreaks, Mexico did not declare COVID-19 as an epidemic and public health emergency until 23 and 30 March, respectively [2]. The first case was registered in Mexico on 27 February 2020 [3], and the first death associated with the disease occurred on 19 March [4]. The Secretariat of Public Education (*Secretaría de Educación Pública*, SEP) in Mexico officially stated on 14 March that the Easter break would be prolonged from 20 March to 17 April. Even though classes were planned to resume on 20 April, the closure of schools lasted for more than a year due to the unexpected increase in COVID-19 infection cases. In addition, there was no comprehensive national plan to address the impact of the pandemic on the higher education sector, apart from establishing health protocols and

providing suggestions for best practices [5]. As a result, Higher Education Institutions (HEIs) took it upon themselves to develop contingency measures to alleviate the adverse effects of the pandemic on students' academic education.

To the best of our knowledge, no existing review specifically for Mexico or Latin America compiles and analyzes published studies on the challenges encountered by higher education institutions during the pandemic, along with the lessons learned and opportunities for enhancement resulting from this global crisis.

This article aims to present and discuss the effects and impacts of the closure of schools due to the COVID-19 pandemic on higher education in Mexico. Likewise, we sought to analyze the strategies implemented by these institutions to identify the positive aspects and determine the areas that require improvement.

### 1.1. Structure of the Mexican Educational System

Higher education is crucial for developing advanced competencies and knowledge for modern economies. With this academic level, students acquire technical, professional, and disciplinary knowledge and cross-cutting competencies that qualify them for various occupations.

In understanding higher education in Mexico, it is relevant to consider its relationship to the preceding educational cycles. The Mexican educational system follows the International Standard Classification of Education (ISCED) and includes three compulsory levels for primary, secondary, and high school education (ISCED 1, 2, and 3, respectively). Post-secondary non-tertiary education programs (ISCED 4) do not exist in Mexico. The upper tier comprises university technician or associate professional (ISCED 5), Bachelor's degree (ISCED 6), and postgraduate programs (ISCED 7), which include specialization programs that last for one year and master's degree programs that last for two years. The final tier is reserved for doctorate programs (ISCED 8), which require completion of ISCED 7. It is worth noting that all these levels (5–8) are voluntary.

The proportion of 25–34-year-olds without upper secondary education in Mexico decreased from 65% to 50% between 2008 and 2018; it remains significantly higher than the OECD (Organization for Economic Cooperation and Development) average of 15%, as reported in 2019 [6]. These values highlight a concerning disparity in the transition from secondary school to higher education.

The higher education system in Mexico is robust and diverse, with 8539 institutions, including 2997 public and 5542 private schools [7]. Enrollment for the 2021–2022 school year reached 5,069,111 students (53.5% female; 64.1% in public universities), making it the second-highest registration among OECD members after the United States. Only 0.1% of people aged 25 to 64 in Mexico have a doctorate, the lowest value among OECD countries.

This complex system comprises 13 subsystems organized according to a government agency, funding source, specialization, or program level [8]. The country's higher education system has two types of subsystems: public and private (Figure 1).

The most important public institution of higher education in Mexico is the National Autonomous University of Mexico (*Universidad Nacional Autónoma de México*, UNAM). It is regarded as the most prestigious university in Mexico and the first in America [9]. It is currently ranked 104th in the Quacquarelli Symonds World University Rankings 2023 [10]. The Monterrey Institute of Technology and Higher Education (*Instituto Tecnológico y de Estudios Superiores de Monterrey*, ITESM), commonly known as *Tecnologico de Monterrey,* is the highest ranked among private institutions, at position 170 in the same rankings.

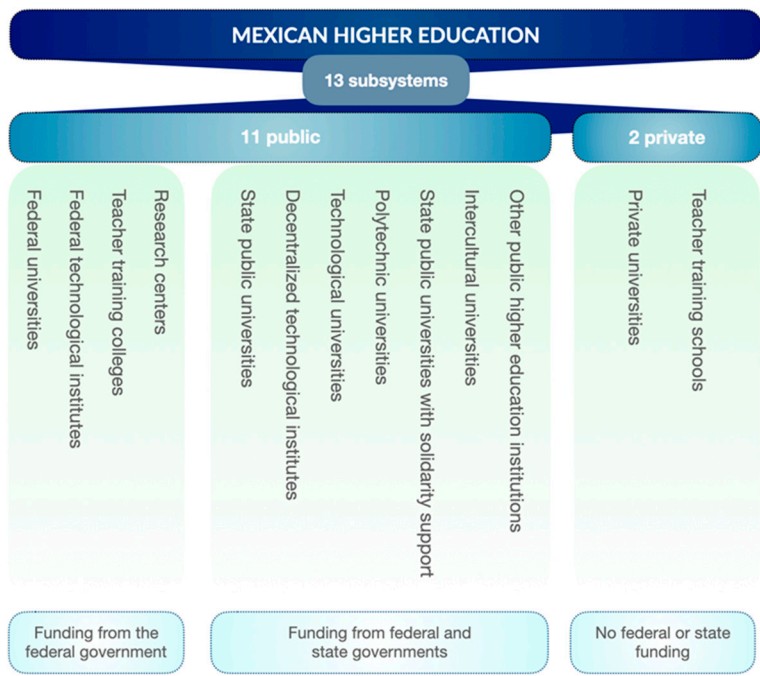

**Figure 1.** Mexican Higher Education subsystems.

### 1.2. Inequalities in Education in Mexico

The right to professional education is essential to a nation's economic and social advancement, and it is encouraging that its accessibility has expanded worldwide in the last two decades. However, some obstacles still impede equal access to this academic level, such as poverty, discrimination, entrance exams, and high tuition fees, among other factors [11]. These severe challenges persist in Mexico as well.

The Mexican higher education system encounters significant hurdles in maintaining quality and preparing students with relevant competencies for the job market. Although Mexico's wage premium for workers with higher education is among the highest in the OECD, the country also has the most significant gender gap. Men with this academic level can earn 78% more than those with only upper secondary education, while women can only expect to reach 66% more [8]. However, only 56% of Mexicans believe everyone has access to education regardless of economic status. Even if access were universal, the quality of education is not guaranteed, with Mexican pupils performing poorly in reading, mathematics, and science compared to their peers in other OECD nations. Obstacles such as poverty, gender inequality, indigeneity, and budget reductions curtail the opportunity to receive an education that meets high standards.

#### 1.2.1. Poverty

In Mexico, there is a notable disparity between the wealthy and the impoverished, with nearly half the population experiencing economic hardship. At the same time, approximately 18% of the country's residents face extremely challenging living conditions. This situation restricts the right to personal growth and education as poverty impacts the likelihood of attending school and the standard of education one receives. Unlike private universities, whose tuition fees have a high cost associated with their prestige, in public universities, the tuition varies from a symbolic payment to an approximate price of USD 40 per year. To help students of low socioeconomic status and motivate them to persevere in their studies in public and private HEIs, the Federal Government applied some strategies: (1) the National Scholarship Program for Higher Education; (2) a financial program to support private banks that grant educational loans to study in high-quality

private universities, and (3) other economic programs funded by the National Council of Science and Technology (*Consejo Nacional de Ciencia y Tecnología*, CONACYT) [12].

### 1.2.2. Gender Inequality

Gender inequalities in education are much less acute than those derived from a socioeconomic origin, where the figures in education have improved in the last 30 years. In many PISA-affiliated countries (Program for International Student Assessment), although young men tend to do better in math and science, women outperform them in language. Some differences do not vary over time; therefore, it can be considered a "scenario of continuous inequalities" regarding gender. In addition, girls were more likely than boys to drop out of school by 12 due to domestic activities and cultural issues, and women were less likely than men to enter and graduate from university. This trend is observed even in career choice. Women tend to pursue careers in care-related fields such as education and healthcare. In contrast, men tend to focus more on disciplines related to the exact sciences, natural sciences, and engineering [13].

Completing higher education can bring about greater advantages for women. In 2017, 72.2% of young women (aged 25–34) with a professional degree were employed, significantly higher than those with only upper secondary education, where the employment rate was 54.3% [8]. Their employment rate is also lower (74.2%) compared to 87.9% among men. This disparity can be attributed to discriminatory practices in the business sector, particularly against women who are also young mothers.

### 1.2.3. Indigeneity

The great socioeconomic inequality that has prevailed in about half of the inhabitants, together with ethnic diversity, are linked to disparities in access, permanence, and completion of the different types and levels of schooling.

Nearly 10% of the Mexican population identifies as indigenous, and many speak autochthonous languages, contributing to the country's cultural diversity. Most native people (64%) live in rural areas and rely more on natural resources. However, they are more vulnerable due to inadequate housing and limited access to healthcare. In addition, they are more likely to live in poverty (approximately 75% of them were considered poor in 2012), almost twice as likely as the non-indigenous population [12]. Consequently, their achievement and employment rates are significantly lower than non-indigenous groups. The pandemic has exacerbated this situation [14].

### 1.2.4. Reduction of Educational Budgets

One of the challenges facing the educational system in Mexico is the reduction of educational budgets, with Mexico's annual spending per student below the OECD average. Mexico's public expenditure on all levels of education is the second highest among OECD countries (16.4% of total government spending in contrast to 10.8%). Still, spending per student remains the lowest (USD 3600 per year on average across all levels, USD 5900 at the higher education level). From 2008–2018, although the higher education level increased from 16% to 23%, it was still below the OECD average of 44% [8]. This expansion has been mainly due to a series of government policies, increased geographic coverage, and a strengthening of the distance learning system. The Mexican government further cut the education budget by more than 11% in 2017 due to economic pressures, which resulted in a 40% reduction in teacher training programs and a third cut in the textbook budget, exacerbating an existing problem.

### *1.3. Distance Education and Internationalization*

Distance education has played an essential role in broadening access to higher education. This approach to teaching involves students being able to study at their own pace without the requirement of being physically present at the institution where they are

enrolled [15]. This educational model was initially implemented in Mexico to promote literacy outreach in rural communities.

In the past 30 years, there has been a growth in both public and private institutes offering distance education programs in Mexico. During the implementation of distance education in Mexico, it was found that institutional collaboration is a crucial strategy, and promoting social awareness is vital to prevent further disparities.

Internationalization is a goal of the distance education program. According to Camacho Lizárraga [16], the most effective internationalization strategies for Mexican HEIs are summer exchange and dual degree programs. However, the COVID-19 pandemic and its associated travel restrictions and limitations on face-to-face attendance have prompted HEIs worldwide to adopt an "internationalization at home" approach. This approach involves co-curricular programs, online collaboration with foreign universities, research publications in international journals, research publications in international journals, participation in research teams from different countries, and offering international seminars. This shift can potentially increase the number of Mexican higher education students with at least one international experience, which currently stands at around 1%, compared to the average of 2% in most OECD countries [8].

## 2. Methodology

A comprehensive literature search was conducted across various academic databases, including WoS (Web of Science), SCOPUS, SciELO (Scientific Electronic Library Online), Google Scholar, and Redalyc (The Network of Scientific Journals of Latin America and the Caribbean, Spain, and Portugal). The search utilized a set of predetermined keywords to capture relevant studies published between March 2020 and February 2023 related to higher education in the context of the COVID-19 pandemic, explicitly focusing on Mexico. The keywords used in English were "Higher education" OR university OR undergraduate OR graduate OR HEI OR "Tertiary education" AND "COVID*" OR "pandemic" AND Mexic*. In addition, a separate bibliographic search was conducted using corresponding Spanish keywords: "Educación superior" OR universidad OR licenciatura OR postgrado OR IES OR "Educación terciaria" AND "COVID*" OR "pandemia" AND Mexic*. As most of the publications focused on the country's central region, the search was expanded by incorporating the names of additional states within the Republic as keywords. This was performed to encompass a broader geographical scope and include a more significant portion of the country. Furthermore, national databases such as INEGI, SEP, ANUIES, Secretaría de Salud, and relevant newspaper sources were consulted to provide contextual statistical data.

A total of sixty-nine articles were reviewed due to this rigorous selection process. Peer-reviewed articles and book chapters were considered for inclusion, whereas publications primarily based on interviews and personal comments were excluded. The collected information was carefully examined, identifying two main central themes: (1) Impacts and implications on Higher Education and (2) Innovation in teaching and learning experiences. Regarding the first theme, the studies concerning the effects of the pandemic on higher education were categorized into the following subthemes: (a) Connectivity and digitization, (b) Effect on emotional and psychological well-being, (c) Student perception, (d) Faculty and academic staff perception, and (e) Adaptation to change. On the other hand, the second theme focused on the following subthemes: (a) Educational platforms and digital learning technologies; (b) Teaching strategies such as gamification and the flipped classroom; (c) Online laboratories; (d) Simulations, virtual reality, and augmented reality; and (e) International collaborative learning.

## 3. Teaching and Learning Experiences in Mexico during the Pandemic
### 3.1. National and Local Government Policy Measures to Face COVID-19

The Mexican Secretary of Health implemented several community mitigation strategies recommended by the WHO, such as quarantine, social distancing, and the cessation

of non-essential activities, including in-person classes [17]. Despite these actions, the government received intense criticism for its slow response. The University of Celaya (*Universidad de Celaya*, UDEC) in Guanajuato was the first university in Mexico to halt in-person activities, sending learners home on 13 March. In addition, 14 out of 20 ANUIES universities in Mexico City had to cancel classes by 17 March [18].

Niño Carrasco et al. [19] analyzed the responses of 35 autonomous public universities in Mexico. They found that eighteen of them had emergency plans to protect the well-being of all community members, including academic, physical, and psychological aspects. Furthermore, seven universities provided teacher training remotely while ensuring the quality and continuity of the educational program. Six institutions offered training for designing online classes, and nine provided technical support to students [5]. Regarding admission processes, UNAM and the National Polytechnic Institute (*Instituto Politécnico Nacional*, IPN) conducted face-to-face exams while maintaining a safe distance between students and using masks and transparent protectors to minimize contact. In contrast, the Autonomous Metropolitan University (*Universidad Autónoma Metropolitana*, UAM) and the Autonomous University of the State of Mexico (*Universidad Autónoma del Estado de México*, UAEM) opted for online exams [20].

### 3.2. Impacts and Implications on Higher Education

Several significant challenges were identified during the pandemic's early stages in Mexico's higher education. These included the lack of a technological infrastructure that could cater to the needs of all students, the need for teacher training in emergency remote education and managing a heavy workload and time effectively, the digital gap, the limited resources of trainees, and the lack of assessment tools in virtual teaching contexts [21].

The closure of schools to prevent the virus's spread resulted in a shift from traditional in-person classes to online instruction, with the subsequent digital transformation. While this resolution proved successful for some schools, it could be argued that the demands of coursework had to be balanced against the needs of families, such as restricted space and access to digital devices and connectivity for attending classes. This situation led to a critical increase in internet connections in the country. In 2019, a significant proportion of Mexican households, amounting to 44.2%, had access to Internet services. However, this figure increased in subsequent years due to the pandemic, to 59.9% in 2020 and 66.4% in 2021, according to preliminary data [22].

Moreover, an increase in the percentage of individuals who used computers as a tool for school support was observed, rising from 44.5% in 2019 to 51.5% in 2020. Students also reported increased costs during confinement due to studying and working from home [23]. These costs were mainly due to increased electricity consumption, the purchase of electronic devices, improved internet access, and mobile data usage).

Although learners and teachers had to acquire technological skills rapidly, the pre-existing infrastructure and technical capacity were crucial. The responses of the HEIs were influenced by their prior experience with remote classroom systems, which was a significant factor. A study reported that before the pandemic, 65% of students at private universities affirmed had a virtual campus and obtained better results when evaluated at a distance [24]. However, students at public universities were more familiar with educational platforms (77%) and distance education (64%) and had taken online classes previously (42%). The students from private universities had better technological equipment (computers, tablets, etc.) and internet connections, which greatly aided in the transition to remote learning. Similar research [25] revealed that instructors from private universities reported that their institutions had better infrastructure conditions, virtual campuses, and online course offerings even before the health emergency. They also mentioned being more acquainted with educational platforms, conducting remote student evaluations, and providing online educational opportunities. Conversely, public university teachers were more familiar with distance learning tools and were better positioned to achieve online training courses. A

survey at UNAM [26] revealed that teachers faced logistical and technological challenges, followed by pedagogical issues, while the socio-affective situation was less concerned.

The consequences for the academic and administrative staff included budget cuts for education to redirect funds to health and social welfare, affecting workers' salaries. In institutions such as UAM, a decrease of 11.8% in the education budget was observed between the second semester of 2019 and the first semester of 2020 [20]. Some teachers' temporary contracts were terminated at IPN due to unclear strategies for continuing distance teaching. It was also observed that specific educational programs were better adapted to online education than others. Specifically, engineering and technology programs were found to be more thoroughly adapted, while social education programs lacked the necessary training in technological platforms and learning management systems (LMS). Reductions in university budgets led to administrative layoffs, which affected the teaching-learning process [27].

The lack of sufficient time for teachers to prepare themselves and receive training resulted in students relying heavily on self-directed learning, primarily due to the absence of social interaction. Despite the efforts of educational institutions to provide alternative means of education, some students, especially those with low incomes and disabilities, experienced limited learning opportunities. Many had to share devices and deal with slow or unreliable internet speeds. For some, purchasing additional internet plans on their phones has been necessary [28]. As a result, the Mexican attrition values in higher education (excluding postgraduate) increased from 7.0% in 2015–2016 to 8.4% in 2019–2020, reaching a peak the following year (8.8%)). Sinaloa and Quintana Roo were the States most affected, with their dropout rates rising from 8.8% to 18.0% and 14.3% to 18.0%, respectively, between 2019 and 2020 [7]. This trend was observed worldwide [29,30], even in primary and secondary education [31]. These reports analyze various indicators such as school closures, student enrollment, dropouts, and student perceptions. Additionally, experts have made predictions about the likely impact of the pandemic, drawing on past instances of instructional disruption and extrapolating from them. For example, Hallgarten [32] identified factors such as reduced access to educational services, school closures, poor quality education, fear of returning to school, and emotional stress as the primary reasons students dropped out during the COVID-19 pandemic.

The transition to online learning has also had personal implications for scholars [24]. The emotional impact of online education is still being studied [33], with mixed reactions, but some have praised teachers for their empathy. Finally, students faced unique challenges, such as caring for sick or elderly family members while keeping up with academic responsibilities [34]. A series of reports in the literature highlighting cases from the Mexican territory has been compiled and organized into five themes: (1) Connectivity and digitalization, (2) Effect on emotional and psychological stability, (3) Perception of students, (4) Perception of teachers and academic staff, and (5) Adaptation to change.

### 3.2.1. Connectivity and Digitalization

The transition to online learning posed three major challenges: limited access to technology, limited mastery of technology, and the need for pedagogical reform to adapt to distance learning [35]. Many educational centers' lack of LMS also posed a significant obstacle to students seeking to continue their studies during the pandemic. In Mexico, only 39% of higher education institutions offered LMS support, with the majority using Moodle (71%). Additionally, 87% of these schools needed more qualified IT personnel, and only 9% had a mobile app to provide students with administrative and academic access. Consequently, IT played a minimal role instead of a strategic one during the pandemic, providing inadequate infrastructure and failing to enable innovative educational delivery methods [35].

In general, the most successful institutions have been those with extensive prior experience in online teaching. Some private schools, such as *Tecnologico de Monterrey*, provided many resources to equip their facilities with appropriate tools, such as in-classroom cam-

eras and Zoom accounts for each teacher. Moreover, they provided constant training to their professors in using online teaching tools. This support was also reflected in furniture, tablets, or laptops for teachers to equip themselves adequately. *Tecnologico de Monterrey* was the first Mexican HEI to move its educational offerings from 17 March by the "flexible and digital model," through which it has provided 45,000 synchronous weekly classes for 90,000 students [36]. This university trained its 10,000 teachers in "active learning in a digital context," evaluation strategies, and using digital platforms [5].

Public universities showed a completely different picture. For example, 20% of the IPN undergraduates did not have online classes, while 10% did not take courses in that format [37]. UNAM has shown the complex reality of online learning. Only in the main campus called *Ciudad Universitaria*, located in Mexico City, 33% of its more than 130,000 students did not have internet services. Approximately 10% did not have Internet or computer service [38].

Even with this unfavorable panorama, these universities have made an extraordinary effort to maintain the quality of teaching that characterizes them. UAM was one of the leading public institutions in Mexico to distribute computer equipment with an Internet connection to low-income pupils. As of May 2020, the university announced that it had delivered 4324 tablets with Internet access as part of the Emerging Remote Teaching Project (*Proyecto Emergente de Enseñanza Remota*, PEER) [5]. UNAM, through the campaign

"UNAM doesn't stop" ("*La UNAM no se detiene*") [39,40], created different bodies to coordinate the institutional response to the virus. This operation included the University Commission for the Attention of the Coronavirus Emergency, the COVID-19 Diagnostic Center, and the Legal Observatory of the Pandemic. UNAM undertook actions to reduce the digital divide through the PC Puma program. These actions included installing 14 computer centers for students in the Valley of Mexico, 58 with 4000 computers and high-speed Internet, acquiring 25,000 tablets with an Internet connection they can take home, and allocating 12,000 connectivity scholarships [41].

The situation experienced in higher education depended greatly on the possibilities of each state, where technological, social, and economic inequality is present in different state sectors. For example, Chiapas is one of the four poorest states in Mexico, according to the National Council for the Evaluation of Social Development Policy [42]. Ochoa mainly analyzed aspects of connectivity, digitalization, and the conditions of students during online education at their homes at the Autonomous University of Chiapas (*Universidad Autónoma de Chiapas*, UNACH) [43]. The university professors decided to maintain more active communication using resources such as Facebook, WhatsApp, and email while managing academic work resources, such as the Educa-te platform and synchronous work resources: Google and Zoom were limited.

Similarly, the Pedagogical University of Veracruz (*Universidad Pedagógica Veracruzana*, UVP) responded quickly to the pandemic by implementing emergency measures to switch from face-to-face to virtual instruction. They included online activities and electronic platforms such as Zoom, Google Classroom, and Edmodo. However, due to time constraints, they adapted only the classroom content to electronic media without a proper teaching approach. The university used Microsoft Teams as its institutional platform, facilitating the formalization of virtual academic activities [44].

### 3.2.2. Effect on Emotional and Psychological Stability

It is noteworthy to examine how a broad spectrum of emotions such as anxiety and stress has affected individuals in academic communities, the resulting outcomes that have emerged since the outbreak, and the subsequent implementation of lockdown measures [24,45]. In this context, Valencia et al. [46] researched the impact of COVID-19 on Mexican citizens in terms of Clinically Significant Depressive Symptomatology (CSDS) in 2021. The study revealed that experiencing symptoms of depression was more prevalent among individuals with lower levels of education, which is a significant discovery considering the high rates of non-enrollment in Mexico's education system. Within the

psychological consequences, an increase in negative feelings, such as fear, anger, frustration, despair, loneliness, and symptoms of anguish, anxiety, and depression, have been reported. Likewise, ref. [47] discovered that nearly half of the Mexican adults experienced anxiety during confinement. In particular, ref. [48] found that undergraduates in Mexico were also affected, with high levels of anxiety, stress, and depression. Burnout and high emotional exhaustion were identified in first-year medical undergraduates.

Similarly, the teachers also faced challenges to overcome. Delgado-Gallegos et al. [49] applied a revised edition of the adapted COVID-19 stress scales (ACSS) that included teaching anxiety, preparation, and resilience for academic professionals in Mexico. The statistical findings indicated a significant correlation between resilience, teaching anxiety, and preparedness, and with all areas of the ACSS scale (danger, contamination, social-economical, xenophobia, traumatic stress, and compulsive checking). Academic professionals have also demonstrated high levels of resilience during the quarantine, likely due to their ability to adapt to the stress brought on by the transformation of the educational system. Another research study examined how the pandemic has affected professors' sleeping patterns and sleep quality at various universities in Mexico [50]. According to this report, the time individuals slept on weekends has decreased by more than an hour, possibly because emotional exhaustion, depersonalization, and lack of professional fulfillment affect sleep quality. The authors suggest two possible factors contributing to this trend: increased academic workload in distance learning centers and the need for training in technological tools. On the other hand, it is possible that the reduction of social jetlag (the difference between weekend and weekday sleep schedules) helped to meet teachers' sleep needs better.

Fortunately, not all effects and emotions observed due to the pandemic have been adverse. As reported by Gaeta et al., most students from various universities in Mexico stated experiencing positive emotions such as gratitude and joy despite facing challenging circumstances [51]. Although anxiety and frustration have been prevalent, specific individuals also experienced happiness, confidence, and optimism. It is worth noting that the participants utilized more problem-focused coping strategies than emotion-focused ones, including constructive reappraisal and seeking social help. Moreover, the student's capacity to handle the situation and the support from their family and social networks contributed to these beneficial emotional experiences.

### 3.2.3. Perception of Students

McClelland's Acquired Needs Motivation Theory says humans have three emotional needs: achievement, power, and affiliation [52]. A recent report examined the issue of belongingness among STEM undergraduates at private universities in Mexico and the USA [53]. The study found that they felt disconnected due to the lack of practical laboratory experience and the reliance on theoretical demonstrations in online classes. While the technology did not hinder them, they struggled with building meaningful connections with their peers and professors, impacting their ability to collaborate effectively on assignments. These communication barriers had a detrimental effect on their motivation to participate actively in class discussions. In addition, students reported difficulty concentrating during online classes and experiencing fatigue from extended screen time.

In the same line of thought, several studies have focused on the perception and motivation of pupils to pursue their careers during the pandemic. Researchers have been particularly interested in exploring this topic, as evidenced by the investigation into undergraduate and graduate students at the UVP [44]. They employed self-management and organizational strategies based on the UVP's "*Horizonte Educativo*" (Educational Horizon) model, which emphasizes developing divergent thinking by learning how to learn. This approach incorporates transversal skills such as self-organization, autonomy, creativity, communication, language, collaborative work, ethics, and Information and Communication Technologies (ICTs). Students prioritized their activities by creating schedules that included both academic and non-academic pursuits. They communicated with their classmates and teachers through platforms such as WhatsApp or Skype, engaged in autonomous

learning, diversified their activities to share resources with others in their households, and approached their work as if they were in a face-to-face environment. Their primary motivations were personal growth, financial stability, education completion, and family support.

In a study held in the state of Yucatan, located in southeastern Mexico, female undergraduate students attending private institutions have remarked that despite using various communication platforms such as WhatsApp, Zoom, Google Meet, House Party, Skype, and Facetime to stay in touch, they had felt that communication is not entirely effective and often leads to misinterpretation [54]. Furthermore, they have expressed concerns about not being able to see their friends in person and struggling to comprehend their course material. In contrast, female students attending public institutions in the same state primarily worry about a lack of resources to support their studies, such as poor internet connections and inadequate computers. However, they appear to have more social cohesion, receiving more significant support from their friends via social networks and platforms, making social distancing more manageable. Both groups of students have faced similar challenges, including excessive homework, boredom, fatigue, internet or power outages, difficulty grasping the subject matter, getting easily distracted, lack of flexibility and patience from teachers, and the unsuitability of their homes as learning environments. The learners have noted that many teachers lacked the necessary training and resources to deliver their classes through educational platforms, resulting in project and task-based learning that covered the content. Additionally, the study revealed that private school students had noticed a more significant involvement of parents, who often attend classes alongside their daughters, providing better attention from educational authorities.

Regardless of their geographic location, students' viewpoints in Mexico are similar. In the urban region of Oaxaca, one of Mexico's poorest states in the south, public and private school students expressed discomfort with the mandatory switch from in-person to virtual classes [55]. Some of them disagreed with this transition due to inadequate communication with their teachers, an excessive workload without prior explanation or feedback, and in some instances, difficulties with connectivity that impeded their comprehension. In contrast, a minority who felt comfortable with remote learning cited the advantages of a peaceful learning environment and time savings. The primary barriers faced by students included communication issues, limited internet access, and disorganization.

Students enrolled in secondary and higher education programs in Sonora, a northern state in Mexico, encountered various challenges while engaging in remote learning [56]. These obstacles included limited access to technological resources, insufficient commitment and responsibility when completing academic tasks, domestic issues, and a lack of proficiency in independent study skills. To ensure that classes continued uninterrupted, students identified three key strategies: implementing online learning platforms, transmitting school assignments via email or other channels, and participating in the "Learn at Home" program. Most respondents reported that communication between them and their instructors was satisfactory. Despite this positive feedback, some noted missing social interaction and community-building opportunities in face-to-face learning environments.

An intriguing observation was made in independent research, which linked adaptability to academic achievement [57]. The students who performed well academically also had a favorable perception of the online teaching they received, even if it was not their preferred method of instruction. Conversely, pupils who received low grades in their courses evaluated the university's remote education and transition to the "New Normal" more negatively.

### 3.2.4. Perception of Teachers and Academic Staff

The perceptions and emotions of educators concerning their experiences with online classes have been subject to analysis. In a study by Zamora-Antuñano et al. [58], teachers' opinions regarding virtual learning platforms, such as Moodle, Google Classroom, and Blackboard, were investigated during the transition from traditional teaching methods to emergency distance learning. The study found no significant differences in platform

preferences based on gender, level of education, teaching experience, or whether the teacher worked in a public or private school. However, teachers in private schools reported better proficiency in technology management than their public school counterparts. The report also revealed that teachers' confidence in technology management decreased as they grew older. The primary challenges faced by teachers during the pandemic were internet connectivity, student absenteeism, and students' receptiveness to online learning platforms.

Digital educommunication media (DEM) was essential in transitioning from in-person to online classes. Previous familiarity with online resources was critical in effectively integrating digital tools. Rey-Ronquillo and Machin-Mastromatteo [59] found that 75% of professors from two Mexican institutions had prior experience with virtual education. Throughout the pandemic, most professors in the study (80%) used DEM, such as email, images, animations, video, and cloud-based files. Some negative aspects included limited personal interaction, student disengagement, and technical difficulties. Despite this, professors believed that remote learning would not entirely replace face-to-face models, but having skilled communicators as instructors would benefit both approaches.

Other studies also highlight the benefits and challenges that the participants had. Most teachers surveyed from universities in Oaxaca expressed positivity towards working comfortably from home, viewing it as an opportunity to enhance their teaching skills [55]. They echoed their students' concerns about limited internet access, poor communication, and lack of interaction. The study also discovered that instructors experienced overwhelming challenges in two distinct areas: structural conditions (including access to technology and internet connectivity) and personal limitations (such as socioemotional competencies, adapting to new learning methods, digital skills, communication, and organization). Educators agreed that shortly, it would be crucial to promote self-directed learning, autonomy, and socio-emotional competencies in students. On the other hand, teachers from Sonora have pointed out that they took several measures to aid students in adapting to remote learning through institutional procedures [56]. These include providing courses and guidance to educational staff, setting up software and virtual platforms to ensure uninterrupted classes, and creating manuals to assist teaching groups during the health crisis, among other initiatives.

### 3.2.5. Adaptation to Change

Gutiérrez-Salomón et al. conducted a study to explore teachers' experiences regarding the forced integration of technological tools resulting from the pandemic [60]. They identified three stages: accelerated transition, adaptation to electronic devices, and technology ownership. The study revealed that the shift to online teaching practices (first phase) exposed deficiencies in technological literacy. During the second stage, the professors underwent arduous training, resulting in increased self-confidence and improved working conditions at home for teaching purposes. In the third phase, significant changes were observed in attitude, integration of computers, and problem-solving resulting from technology. The authors concluded that having the specialized equipment is insufficient; the actors involved in the educational process must develop competencies in using and managing ICTs gradually and systematically. It is also necessary to create a forward-looking disposition towards using computers and propose literacy where ICTs are used, strengthening the cooperative work of students and teachers. Furthermore, the new teaching modality during confinement provided an opportunity to listen to students, hear their struggles and ways to overcome the situation and facilitate the collective construction of identities of resistance.

The response of universities to this accelerated transition to online learning has been classified as either reactive or proactive [34]. Reactive measures included using established LMS or allowing educators to design alternatives. Proactive measures involve building a high-quality virtualization platform that integrates the institution's psycho-pedagogical, techno-pedagogical, and educational principles. In Oaxaca, for example, teachers and students in public and private universities initially showed significant resistance to change

when faced with school closures during the health emergency. However, implementing digital tools such as Meet and Classroom helped mitigate the problem. The main obstacle was providing high-quality services and developing strategies to address issues arising from the digital divide and students' socioeconomic status [55].

Realyvásquez-Vargas et al. [61] examined the impact of environmental factors, including lighting, noise, and temperature, on the academic performance of online students from 206 universities in Mexico during confinement. The findings indicate that these factors directly impacted their intellectual development, concentration, and comfort while studying from home. The authors argue that academic success is not solely dependent on the amount of time spent studying but is also influenced by environmental conditions. Therefore, HEIs should create a pleasant learning environment, including appropriate lighting, noise levels, and temperature, to benefit students in face-to-face teaching situations. For online learning, the authors recommend that teachers use more interactive approaches and avoid assigning homework that may increase students' screen time. Additionally, pupils should have access to a private space to minimize distractions. The study area should be ergonomically designed, with temperature control features to enhance sustainability for both students and HEIs.

### 3.3. Innovation in Teaching and Learning Experiences

Over the years, HEIs have acknowledged the demand for innovative educational models that cater to the current needs of a technologically advanced and globally connected society. The COVID-19 pandemic has presented a significant challenge for education in general and specifically for higher education. However, it has also acted as a catalyst to accelerate the implementation of innovative strategies that address these challenges. These responses have varied from temporary and reactive measures to long-term visionary approaches.

The pandemic undoubtedly offered a unique chance to create new and innovative projects that could revolutionize learning processes and influence academic institutions. Today, it is essential to consider whether the SARS-CoV-2 virus disruption served, in addition to demonstrating the natural resilience of HEIs, to present an opportunity for innovation and a learning experience.

All the while the pandemic, Mexican HEIs strengthened five primary resources, which included (1) the utilization of educational platforms and digital technologies; (2) implementing didactic strategies such as gamification, problem-based learning, and flipped classrooms; (3) establishing online laboratories; (4) incorporating simulations, virtual reality, and augmented reality; and (5) participating in international collaborative learning.

### 3.3.1. Educational Platforms and Digital Learning Technologies

During the transition to online learning, the use of educational platforms became more widespread among institutions. These websites offer online classrooms, video lectures, interactive quizzes, and communication tools that allow students and teachers to collaborate and communicate remotely. They have also helped to bridge the gap between students and teachers by providing a virtual classroom environment where they can interact and engage with each other in real-time. Many offer features such as online chat, video conferencing, and collaborative whiteboards that allow students to ask questions and receive feedback from their teachers and peers. The most used platforms during contingency were Zoom, Google Classroom, Canvas, Moodle, Edmodo, and Kahoot. Using these websites has been fundamental to ensure that learning continues despite the disruption caused by COVID-19. An example of concrete actions taken in Mexican HEIs regarding digital platforms is the *Universidad Autónoma del Estado de Hidalgo* (UAEH), which trained its professors in using the Garza platform, supported by Moodle. Thus, they could upload all the necessary class information [62]. The *Universidad Autónoma de San Luis Potosí* designed subjects in virtual learning environments using an LMS under a developmental approach. They included three critical moments: (a) diagnosing prior knowledge, (b) using an inverted classroom

to discuss the concepts learned in class, and (c) recapitulating everything reviewed in class [63].

The "New Normal" led to the promotion of previously unexplored competencies based on students' personalities or academic profiles. For instance, in a physics course at *Tecnologico de Monterrey*, about 30% of undergraduates used an iPad for taking notes during class. This significant and positive trend revived the practice of taking notes in class, which had decreased in the two years before the pandemic when they only took pictures of the blackboard. Moreover, applications such as Verbe, Socrative, and Desmos have been used to encourage social learning in classroom activities, and smartphones have been utilized as measurement tools, transforming the traditional face-to-face approach [57].

In addition to the educational platforms and digital tools, the availability of online courses through MOOC (Massive Open Online Courses) platforms such as Coursera, EdX, Iversity, XuetangX, and ICourse163 was of great relevance since they enable students to access course materials from anywhere and at any time. Several authors have noted that integrating technology tools creates new virtual environments that enhance learning, motivation, and satisfaction [64,65]. Research has shown that teachers from UNAM believe that the organization of school activities in MOOCs positively influences students' motivation, engagement, and learning [65].

### 3.3.2. Didactic Strategies: Gamification and Flipped Classroom

The gamification approach incorporates game-like elements, such as points, badges, and leaderboards, into non-game contexts to increase engagement and motivation. During the pandemic, gamification has emerged as an effective educational tool to enhance student engagement and learning outcomes in remote and hybrid learning environments.

Some HEIs applied this strategy in Mexico to motivate and engage students by making learning fun and interactive. For example, Chans and Portuguez Castro [66] created a gamification approach for a chemistry course at *Tecnológico de Monterrey,* easily implemented in other settings. The process included achievements that motivated and incentivized learners to accomplish specific goals, categorized according to their short-term, medium-term, or long-term benefits. The gamification approach successfully improved student motivation, engagement, attitudes, and academic performance. Another gamification model used a reward system featuring badges, avatars, and leaderboards for participation and effort [67]. Two undergraduate courses (Calculus and Development of Transversal Competencies) implemented this approach to promote students' attention, motivation, and continuous engagement in an active learning scenario. The model also utilized social networks such as LinkedIn to enhance participants' connections.

The same authors also investigated the impact of a gamification model on students in academic fields such as Engineering, Economics, and Social Sciences. They found that the reward mechanics motivated Engineering undergraduates to engage in higher-quality activities and favor mathematical challenges. At the same time, Economics and Social Sciences pupils were more inclined to participate in class. The study concluded that gamification could be effective in online courses such as MOOCs. However, it may also present technical challenges in administering awards or other gamification elements [68].

The educational model known as the flipped classroom emerged as a concept in the early 1990s. Still, it was not until 2012 that Jonathan Bermann and Aaron Sams coined the term in their book "*Your Classroom: Reach Every Student in Every Class Every Day.*" The basic idea behind this model is to "flip" the traditional educational approach by presenting theoretical concepts as homework through instructional videos or concise readings that students can review at their own pace. In-person class time is then dedicated to practical exercises guided by the teacher. During the pandemic, the flipped classroom approach applied in a Mexican University was considered a viable and effective alternative due to its pedagogical nature, and students generally responded positively regarding learning outcomes [69].

### 3.3.3. Online Laboratories

Online laboratories, also known as virtual laboratories, became famous as an educational strategy during the pandemic. Through this didactic tool, the students could conduct scientific experiments and explore concepts in a remote setting, which has become particularly important as traditional in-person labs have become challenging or impossible to achieve due to the pandemic. Furthermore, numerous professors devised novel methods for conducting laboratory exercises, enabling students to perform them remotely from their own homes. For instance, professors at *Tecnologico de Monterrey* designed laboratory practices that could be performed at home for biotechnology engineering undergraduates in their fourth year [70]. The teachers assembled kits containing essential tools such as 3D-printed mini-centrifuges and safe handling reagents shipped to students' homes. They also provided real-time remote guidance and multimedia resources to assist scholars in conducting the experiments and achieving the intended knowledge and competencies for the course. The authors conducted a survey to assess student opinions on this pandemic-era teaching approach. The majority expressed confidence, interest, curiosity, and concentration, while fewer reported feelings of confusion, despair, and insecurity.

Similarly, the study by Chans et al. [71] reports adopting a chemistry laboratory course to an online mode through a compilation of hands-on experiments with materials or substances of daily use. It also evaluated the learning students achieved through skills acquisition, showing that at least 68% reached a high domain level of competency. Finally, the authors conducted a statistical analysis to evaluate the students' perception of this remote learning adaptation, concluding that most trainees had a positive attitude toward the class and would also recommend taking the course in this modality.

In general, practical work should be aligned with the learning activities to optimize learning in laboratory courses. According to Salinas-Navarro and Garay-Rondero [72], the best way for students to learn is by being exposed to relevant situations related to their activities. This suggestion is crucial in Industrial Engineering, where students must understand real-world applications such as transportation of goods, inventory management, purchasing decision-making, production, and quality control. However, instructors faced different challenges in creating courses that provided practical experience during the pandemic, especially when they did not have access to the necessary equipment and data. Some key points highlighted by the study were as follows:

- Remote interaction and contextual conditions affect both instructors and students;
- Innovative learning strategies are needed to provide engaging experiences;
- Internet-based activities should incorporate real-world situations to enhance the learning experience.

Although virtual laboratories cannot fully substitute for the hands-on experience of physical laboratories, they offered a valuable alternative that helped students continue to learn and explore scientific concepts during the pandemic. They also have the potential to be a valuable tool for education beyond the pandemic, as they offer flexibility and accessibility that can benefit a wide range of learners.

### 3.3.4. Simulations, Virtual Reality, and Augmented Reality

In the QS World University Ranking, published annually by [10], two Latin American universities stand out for promoting educational innovation and technology. UNAM offers the Support Program for Projects to Innovate and Improve Education [73]. *Tecnologico de Monterrey* has two initiatives: the Novus fund and the Novus La Tríada fund, a partnership with two other private universities, the *Pontificia Universidad Católica de Chile* and the *Universidad de Los Andes* [74]. The most common educational technologies implemented by Novus include extended realities (such as virtual and augmented reality and immersive experiences), web platforms, and mobile applications, with artificial intelligence and virtual and remote laboratories mentioned to a lesser extent.

The COVID-19 pandemic has had a significant impact on medical education, including the training of future surgeons. Due to restrictions, many medical students could not gain

hands-on experience with surgical procedures. In the country, the Mexican Academy of Surgery (*Academia Mexicana de Cirugía*, AMC) advised hospitals to postpone non-essential surgeries, limiting the training opportunities for surgical residents. Therefore, UNAM researchers collaborated with pediatric surgeons from *Hospital Infantil de México Federico Gómez* to create virtual immersive operating room simulators (VIORS) for immersive training in laparoscopic procedures for pediatric surgical residents. This tool was a practical option for developing cognitive and psychomotor skills in laparoscopic surgical practices. The simulator provides a virtual operating room experience that fulfills the educational requirements of residents while complying with social distancing protocols. Moreover, the simulator was easily portable [75].

In the same context, Díaz-Guio et al. [76] evaluated the learning and performance of medical undergraduates in their 4th, 5th, and 6th years at three clinical simulation centers in Colombia, Ecuador, and Mexico. They implemented online-synchronized clinical simulations involving cases related to COVID-19 in both the emergency and operation rooms. The students expressed their fulfillment in learning from this teaching strategy.

Augmented reality is an enhanced, interactive version of a real-world environment achieved through digital visuals, sounds, and other sensory stimuli. Virtual reality, on the other hand, is an immersive experience that helps isolate users from the real world, usually using specialized headsets and headphones. Zamora Antuñano et al. [58] conducted a study at the *Universidad del Valle de Mexico* in Querétaro to examine how augmented reality tools can strengthen students' abilities in Industrial and Systems Engineering programs. Students believed these tools could significantly enhance their acquisition of knowledge, skills, attitudes, and abilities within an educational context. Similarly, the study by Cordero-Guridi et al. [77] describes the creation of a virtual/augmented reality laboratory at the Popular Autonomous University of Puebla State (*Universidad Popular Autónoma del Estado de Puebla*, UPAEP) in Mexico that meets ISO standards for social distancing. The lab was designed to facilitate learning, training, and collaboration among learners interested in additive manufacturing for the automotive industry. A flexible equipment and infrastructure usage strategy was implemented, and positive feedback from students demonstrated the effectiveness of the laboratory for future growth.

### 3.3.5. International Collaborative Learning

During the pandemic, international collaborative learning took on even greater importance. It provided a means for students and educators to connect and learn with peers worldwide despite travel restrictions and lockdowns. Various forms of international collaborative learning were available, such as shared virtual classrooms, joint research projects, online discussions, and student exchanges.

Neria-Piña [78] examined the strategies employed by a Mexican university during the pandemic lockdown. Like many HEIs in Latin America, the university focused on internationalization at home and expanded its scope by increasing the number of students participating in virtual mobility activities. From the results of this study, the institution used technology efficiently to provide multiple virtual international activities, which led to an increase in the number of students with at least one international experience. As a result, the university demonstrated a commitment to diversity, inclusion, and equity in its internationalization efforts.

A highly effective approach to maintaining the advancement of internationalization amidst the pandemic was the implementation of the Global Classroom (GC) strategy. GC is a type of educational method that falls under the COIL (Collaborative Online International Learning) strategy, proposed by Jon Rubin at The State University of New York. This method emphasizes cultivating essential skills such as collaborative work, critical thinking, and problem-solving. The GC program connects professors and students from multiple universities across different countries and cultures, who exchange knowledge to solve a real-world challenge using digital communication tools. According to [79], Mexican students who participated in the GC program were able to develop competencies in sustainability

and effective communication through online interaction with individuals from diverse cultures and disciplines. They also displayed pride in their national identity, respect for other cultures' richness and unique features, and proficiency in utilizing technological tools for communication and distance learning in multicultural virtual environments. Based on the results of this study, COIL strategies such as GC are crucial in fostering global sustainability competencies.

## 4. Lessons Learned and Future Implications

### 4.1. Deepening of the Digital Gap

The pandemic has brought to light unequal access to education due to the digital gap, which varies across different social contexts. This divergence threatened the achievement of the United Nations' Sustainable Development Goal (SDG) 4, which aims to provide equitable and inclusive high-quality education for all. However, the outbreak has also spurred innovation and progress, especially in the digital realm. Two significant educational trends during the contingency were the expansion of distance learning and the increased adoption of innovative educational technologies. The "digital gap" concept has also emerged, referring to society's unequal distribution of ITCs. The gap exists not just among nations but also within particular areas of a country. These inequalities have harmed higher education, and modifications will be necessary in the times ahead.

HEIs faced unique challenges adapting to the post-COVID-19 era due to their varying economic, social, and political circumstances, making implementing a universal strategy challenging [80]. Cooperation is the key to mitigating the pandemic's impact on higher education, and collaboration among nations is necessary to promote positive changes shortly. The digital gap could significantly decrease if developed nations collaborate with emerging and developing countries to offer technical and infrastructural support. Developing countries, such as Mexico, could benefit from e-learning with the appropriate infrastructure, which would help them become a permanent part of the global technology community and society [81].

### 4.2. Benefits of Technology in Education

This crisis positively impacted society's digital transformation. During confinement, people turned to computer technologies to connect with their loved ones and carry out work activities from home. Academic institutions accelerated the adoption and acceptance of technology at an unprecedented rate [81], and the use and learning of technology, which can have long-term benefits in educational design and curriculum development, increased remarkably as well [82]. Online learning has been shown to improve the retention of information and save time, suggesting that the changes brought on by the pandemic could be permanent. Therefore, the effort takes a durability perspective [83].

Teachers and students have learned to adopt a more adaptable, digital, and accountable approach to education. As universities returned to in-person classes in the "new normal" era, they incorporated the knowledge gained from this period. In the coming years, some Mexican institutions will persist in utilizing the online courses they adopted during the pandemic [27]. Advanced technology will probably be used to a greater extent in academic settings, for example, to enhance staff efficiency. This service could include automating administrative tasks, among other things.

Additionally, using big data, artificial intelligence (AI), and statistical analysis can lead to more proactive and effective job performance, revolutionizing how HEIs remain at the forefront of knowledge [83]. Such tools can help with tasks such as determining optimal class sizes, creating curricula, and allocating resources such as facilities and financial aid. In the future, HEIs could incorporate augmented reality and virtual reality tools into labs to supplement in-person interactions between professors and students. Moreover, AI systems could offer guidance and recommendations to scholars regarding course selection and career paths. Unlike conventional methods considering factors such as attendance or grade

point average, AI software can assess student risk based on more detailed information, such as behavior [82].

The COVID-19 pandemic has presented a unique opportunity to transform higher education curricula, prioritizing the needs of students rather than the rote memorization of information. In particular, curriculum changes should emphasize critical thinking, creativity, and collaboration skills, and academic staff should customize them to Bachelor's, Master's, or Doctorate degrees [44]. The outcomes-based education system has been widely implemented to equip graduates with the skills and competencies required in the Industry 4.0 era. Experts advocate focusing on learner-centered instruction and integrating theoretical and practical business learning to cultivate students as future entrepreneurs. Online learning also enables greater curricular flexibility, enabling learners to create learning pathways. In conclusion, significant curriculum modifications that concentrate on practical skills and student development are recommended to prepare them for today's dynamic world.

### 4.3. Limitations of the Study

The limitations associated with this study should be considered within the following contexts: (1) A significant portion of the reports included in our study are in Spanish due to our focus on Mexican universities. (2) A deliberate decision was made not to conduct a systematic search. Instead, we excluded some case references from the metropolitan area of the Mexico Valley (Mexico City and its surroundings) and prominent universities in the country, such as UNAM and ITESM. The purpose was to focus on exploring cases from lesser-known universities nationwide. Therefore, selective searches were carried out to include as keywords the names of Mexican states. (3) Although we have briefly addressed the aspects of emotional and psychological stability in Section 3.2.2, we believe that providing mental health recommendations surpasses this study's scope.

Notwithstanding these constraints, the research makes a valuable contribution to the existing body of literature by providing insights into the influence of the COVID-19 pandemic on academic pursuits. Furthermore, examining articles indexed in different databases provided a comprehensive overview of the existing bibliography. Lastly, incorporating case studies from several states of Mexico allowed us to consider a diverse range of perspectives.

### 4.4. Future Implications and Recommendations

Considering the history and context of HEIs in Mexico and in line with Lytras [27], below are the primary suggestions for reducing the impact of the pandemic and hastening the technological advancement of educational organizations:

- To prevent high-risk students from dropping out, governments should assist them with scholarships, affordable loan options, remedial courses, academic and career counseling, and peer mentoring targeted toward those from low socioeconomic backgrounds. Additionally, authorities should introduce new policies to support different learning formats (face-to-face, online, and hybrid) to enhance the inclusion of students, particularly those from marginalized groups, and encourage innovative educational practices.
- One way to enhance the comprehension of educational practices is to incorporate more digital resources aligned with pedagogical standards (see Sections 3.3.1–3.3.4).
- It is crucial to provide adequate ICT infrastructure and ensure access to the Internet to promote digital skills in the community and enhance the quality of education. Additionally, it is necessary to expand access to ICT resources such as laptops, tablets, and the Internet and ensure that scholars have the required educational conditions at home, including appropriate devices and connectivity, to encourage greater engagement with students inside and outside the classroom (see Section 3.2.1).
- One approach to promote open access and increase the availability of educational resources is to adopt and sustain open-access publishing practices. Additionally,

establishing policies that require all publicly funded teaching and learning resources to be openly licensed for educational purposes is recommended.

- To advance educational research and development, leveraging innovative approaches such as data science, artificial intelligence, and mobile networks to enhance online education applications, methodologies, and tools is essential (see Section 3.3.4). This exploration can contribute to promoting equitable remote learning opportunities that support teaching, learning, and ongoing research.
- One way to facilitate online and hybrid learning is to promote computational and intelligent thinking skills, which can help reduce the cognitive load on individuals. Teachers may better prepare students by fostering such skills to continue learning in these modalities.
- Providing comprehensive training and support to faculty and staff is critical to ensure the effective use of technology in education (see Sections 3.2.1 and 3.2.4). This adjustment includes making significant investments in professional development for teachers, focusing on leveraging technology to enhance their work.
- It is essential to leverage technology, forging strategic partnerships with the private sector, other universities, and the international community, and promoting interdisciplinary collaboration, to foster innovation and collaboration in higher education. By doing so, HEIs can better address emerging challenges and opportunities in today's rapidly evolving world.

## 5. Conclusions

The COVID-19 pandemic has had a significant impact on higher education in Mexico. Although this global emergency has caused many difficulties and disruptions, it has also presented opportunities for reflection and learning. Priorities in the educational field changed prior to and after the pandemic. Before 2020, many HEIs concentrated on sustainable efforts aligned with the United Nations' Sustainable Development Goals (UN-SDGs). However, the COVID-19 pandemic highlighted different economic and social factors, which shifted the priorities of the education sector towards digitalization as a way of attaining sustainability [83].

The biggest challenge encountered by educational systems during digital transformation in crisis contexts was the transition from traditional in-person teaching to online learning, performed without adequate preparation or sufficient experience. Having enough resources, such as human, technological, and organizational resources, was essential to guarantee the sustainability and prosperity of this transition [27].

While online education is undeniably essential, it alone is not enough unless there is a significant shift in educational paradigms [84]. "Teaching" requires thoroughly examining various aspects such as curricula, discipline-focused content, teaching methods, learning and assessment strategies, instructional practices, and academic-administrative management [85]. It is important to note that using ICTs does not have a pedagogical function and may not necessarily lead to innovative pedagogical processes.

Even though the COVID-19 pandemic has brought about numerous challenges, it has also provided educators and political leaders a unique opportunity to reform education systems for future generations actively. As the pandemic continues to evolve, higher education institutions in Mexico continuously grapple with multifaceted consequences, seeking to mitigate the adverse effects while embracing the potential for long-term transformation in pursuing academic excellence. Then, after the crisis, Mexico should focus on enhancing equity, quality, efficiency, and relevance in the educational sector while also developing sustainability plans for the long term based on their specific context in the higher education system.

**Author Contributions:** Conceptualization, G.M.C. and E.P.S.-R.; methodology, A.O.-N., C.O.-N., G.M.C. and E.P.S.-R.; formal analysis, A.O.-N., C.O.-N., G.M.C. and E.P.S.-R.; investigation, A.O.-N., C.O.-N., G.M.C. and E.P.S.-R.; resources, G.M.C. and E.P.S.-R.; data curation, A.O.-N., C.O.-N., G.M.C. and E.P.S.-R.; writing—original draft preparation, A.O.-N., C.O.-N., G.M.C. and E.P.S.-R.; writing—review and editing, G.M.C. and E.P.S.-R.; visualization, A.O.-N., C.O.-N., G.M.C. and E.P.S.-R.; supervision, G.M.C. and E.P.S.-R.; project administration, G.M.C.; funding acquisition, E.P.S.-R. All authors have read and agreed to the published version of the manuscript.

**Funding:** This research received no external funding.

**Institutional Review Board Statement:** Not applicable.

**Informed Consent Statement:** Not applicable.

**Data Availability Statement:** Not applicable.

**Acknowledgments:** The authors would like to acknowledge the financial support of the Writing Lab, Institute for the Future of Education, Tecnologico de Monterrey, Mexico, in the production of this work.

**Conflicts of Interest:** The authors declare no conflict of interest.

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
