# Peer review of "Higher Education in Mexico: The Effects and Consequences of the COVID-19 Pandemic"

_sustainability, doi:10.3390/su15129476_

Round 1

Reviewer 1 Report

Main observations: The topic is interesting, but does not bring new knowledge and the structure of the study is not adequate.

The manuscript consists of a description of the education situation in Mexico and the impact of the pandemic on higher education in the same country. 

It has an introduction chapter and another with the experience of teaching learning during the pandemic, without ever clearly defining the objective of the study or the methodology adopted. It is essential to have a chapter on the methodology adopted. Why the description and the structure followed and not another? Why the sources used and not others? What criteria were used? This information is fundamental to sustain the research.

They state that "A series of reports in the literature highlighting cases from the Mexican territory has been compiled and organised into five themes...". Why and how did the authors select these five themes?

In the abstract the authors begin by stating that "Inequality in student learning worldwide is directly affected by each country's different levels of financial resources, the structure of its government, established educational system, region...". Where is this observation founded if the study does not make a comparative analysis between countries? It would be useful to understand whether the impact of the pandemic in Mexico was similar or not to that of other countries.

On the other hand, the study is descriptive, based on a literature review on the subject. It lacks analysis and interpretation of the results found in the literature. The authors do not add value to the existing literature.

There are also no conclusions. The point "5.3 Future Implications and Recommendations" could be integrated in the conclusions.

I hope that the suggestions pointed out will help the authors to improve the manuscript.

Reviewer 2 Report

Although the topic is not as current as it was a few months ago, it is still important to obtain the results of studies that assess the impact of the covid in educational spaces.

Nothing

Reviewer 3 Report

This chapter describes inequality in education in Mexico  based on
the effects of COVID-19 on educational opportunities.

It is a well-contextualised and well-structured review paper with  clear themes and excellent recommendations.

There are  excellent section summarizing the standard of current offerings in Mexico and the impact of poverty in comparison to OECD countries

It is highly descriptive based on  the research  review from the particular context and the themes (1) Connectivity and digitalization, (2) Effect on emotional and psychological stability, (3) Perception of students, (4) Perception of teachers and academic staff, and (5) Adaptation to change. The themes are balanced and well represented with examples making the paper thorough and most instructive .

The paper is very long so in this reviewer's opinion would help the reader if the authors  connected each of the policy implications in section '';5.3. Future Implications and Recommendation'' with a numbered section in the paper to evidence where each point has been argued.

which include assistance for bachelor’s degree of senior university technician of mothers and heads of households  lines 141/142 make the reader stumber and so rewording  or two shorter sentences might be clearer

I found an odd use of 'transversal 'but maybe it is correct  and a current use of the word- it just jumped out to me as an unusual choice of word for generic skills or graduate skills or similar.

The remainder is excellent with a high level of readability and interest

Reviewer 4 Report

I have enjoyed reading the paper and please see my attached feedback. I would make sure you ask at least 1-2 friends reading the paper thoroughly to make sure it is at a much better level, especially for minor language issues. I have pasted the feedback below as well in case the attached is missed.

......

Title of the manuscript: Learning Opportunities in Higher Education in Mexico: The Effects and Consequences of the COVID-19 Pandemic

Dear researcher(s), you are addressing an important and meaningful gap. Your paper is well-written and it has some important results, and if you edit your paper it can be much more effective. Here some humble suggestions to improve the paper, I would do the following to strengthen the paper. I have enjoyed reading the paper and am looking forward to seeing the paper published. You could increase the effect of your paper with some more recent studies suggested below or any other studies.

Main points:

1.     Title: good and

a.     You may consider shorten the title because more and more journals are asking for manuscripts with less number of total words. If the title is brief, comprehensive the readers and researchers will be more likely to benefit from it more. However, you do not have to shorten it.

2.     Abstract and keywords clear and comprehensive: good and

a.      Change “This chapter..” to this paper

b.     Long sentence, divide into 2-3 sentences

3.     Overall language:

-       The language is quite clear and well-written. You could use an active language for your future papers throughout the paper since an active language seems to be more effective. And more and more researchers go with an active language. However, you do not have to change for this paper- just a suggestion for your future work and I know some journals asking for a passive language.

4.     Length of paragraph : Poor and

a.     Almost ALL paragraphs are too long, you can check the paper and make sure every paragraph is not more than 5 sentences. The best is to stick with 3 to 5 sentences.

b.     Kindly summarize 1.1.3  

5.     Introduction: good and

a.     In the first (and if needed second) paragraph start with how covid-19 has affected all life (e.g., mental health, education, economy) across the world and benefit from some other recent papers. Then start to focus on Mexico in the 2nd or 3rd paragraph. You can benefit from the following papers.

Armiya’u, A. Y., Yildirim, M., Muhammad, A., Tanhan, A., & Young, J. S. (2022). Mental health facilitators and barriers during covid-19 in Nigeria. Journal of Asian and African Studies. https://doi.org/10.1177/00219096221111354

https://scholar.google.com.tr/scholar?hl=en&as_sdt=0%2C5&q=Mental+health+facilitators+and+barriers+during+covid-19+in+Nigeria&btnG=

DoyumÄŸaç, Ä°., Tanhan, A., & Kıymaz, M. S., (2021). Understanding the most important facilitators and barriers for online education during COVID-19 through online photovoice methodology. International Journal of Higher Education, 10(1), 166-190. https://doi.org/10.5430/ijhe.v10n1p166

https://scholar.google.com/scholar?hl=en&as_sdt=0%2C5&q=Understanding+the+most+important++facilitators+and+barriers+for+online+education+during+COVID-19+through+online+photovoice+methodology&btnG=

6.     Thoroughness of the literature review: can be supported with some recent studies yet you do not have to. This will increase the effect of the paper and the journal.

7.     Clarity of the description of the Theoretical Framework (TF):  Not applicable

8.     Research design: Not applicable

9.     Clearly providing research questions and/or purpose: research questions are not applicable, but the purpose of the research is empty.

10.  Choice of research method: Not applicable

11.  Appropriateness of procedures chosen for data collection and analysis: Not applicable

12.  Relevance of data obtained in view of the purpose of the research: well-written

13.  Discussion of the results and their significance: well-written

14.  Soundness of conclusions in relation to data presented: conclusion is empty.  

15.  Limitation:  please provide your limitations (e.g., limited to reviewed/read papers)

16.  Implication:  good and

-       Line 832  “5.1. Deepening of the Digital Gap” this should be “3.1 Deepening of the Digital Gap”

a.     You can increase the effect of your paper by constructing a new section entitled “implication” for clear and brief suggestions in at least two or three of the following most important to mental health, research, administrators, etc.: see suggested paper for implications for specific sections

b.     I would strongly suggest you to call future researchers to use Online Photovoice (OPV) to conduct research on the same or similar topics. The researchers can use OPV, as one of the most recent and effective innovative qualitative research methods. OPV gives opportunities to the participants to express their own experience with as little manipulation as possible if at all, compared to traditional quantitative methods. As researchers one of our responsibilities is to inform others about recent and effective methods, which will increase the effect of your paper and the journal. Future researchers can conduct only qualitative or mixed method to see if OPV. And educators/trainers etc. also can use OPV for experiential activities to increase group and organizational synergy. Please see suggested OPV papers above and the first covid-19 OPV paper below.

Tanhan, A. (2020). Utilizing Online Photovoice (OPV) methodology to address biopsychosocial spiritual economic issues and wellbeing during COVID-19: Adapting OPV to Turkish. Turkish Studies, 15(4), 1029-1086. https://doi.org/10.7827/TurkishStudies.44451

https://scholar.google.com/scholar?hl=en&as_sdt=0%2C5&q=COVID-19+s%C3%BCrecinde+online+seslifoto+%28OSF%29+y%C3%B6ntemiyle+biyopsikososyal++manevi+ve+ekonomik+meseleleri+ve+genel+iyi+olu%C5%9F+d%C3%BCzeyini+ele+almak%3A+OSF%E2%80%99nin+T%C3%BCrk%C3%A7eye+uyarlanmas%C4%B1.+%5BUtilizing+online+photovoice+%28OPV%29+methodology+to+address+biopsychosocial+spiritual+economic+issues+and+wellbeing+during+COVID-19%3A+Adapting+OPV+to+Turkish.%5D&btnG=

17.  Figure/tables: good

18.  In-text reference: well written

19.  References:

-       You could increase the effect of your paper with some more recent studies

-       Please use the following link to include all available doi numbers https://doi.crossref.org/simpleTextQuery simply include your reference one or more than one at a time and submit it. Then you should get all doi numbers if a manuscript has it.

I have enjoyed reading your paper and learned a lot- thanks for your contribution to social sciences. You are addressing an important and meaningful gap. Your paper has some important results, and if you edit your paper based on all or some of the humble suggestions above, it can be much more effective. I am looking forward to seeing the paper published. You could increase the effect of your paper with some more recent studies suggested above or any other studies and not using the suggested ones.

see above and the attached feedback

Round 2

Reviewer 1 Report

I thank the authors for their efforts.

The limitations previously indicated have been overcome.

In my opinion the manuscript is fit to be published.